# How Has the COVID-19 Crisis Transformed Entrepreneurs into Sustainable Leaders?

**Murtaza Haider [1], Randall Shannon [1,*], George P. Moschis [1] and Erkko Autio [2]**

[1] Center for Research on Sustainable Leadership, College of Management, Mahidol University, Bangkok 10400, Thailand; ghandharires@gmail.com (M.H.)
[2] South Kensington Campus, Imperial College London, London SW7 2AZ, UK
[*] Correspondence: a.randall@gmail.com

**Abstract:** EntREsilience, a five-country longitudinal qualitative study, was launched in 2020 in China, Malaysia, the Philippines, Thailand and the UK to understand how entrepreneurs manifested resilience in response to the COVID-19 pandemic crisis events from March 2020 to February 2022. EntREsilience proposed a resilience-manifestation process model describing how entrepreneurs responded to the COVID-19 disruption, aided by external and internal enablers, adjusting their businesses to stabilise and even hunting for opportunities to grow their businesses. The present research adds to the findings of EntREsilience by analysing the strategies applied by entrepreneurs in their response to the crisis. This exploratory study focused on the entrepreneurs' community interactions and studied the effects of these interactions on the response measures adopted by the entrepreneurs. The results describe how the awareness of their stakeholder challenges shaped the entrepreneurial response. Realising the importance of stakeholder well-being to the sustainability of their enterprise motivated the entrepreneurs to develop sustainability competencies towards their stakeholder challenges, innovating solutions for their mutual well-being. By extending the resilience-manifestation process model, this paper proposes a transformation model depicting the process of entrepreneurs transforming into sustainable leaders triggered by stakeholder challenge awareness and moderated by contextual factors.

**Keywords:** COVID-19 pandemic; entrepreneurial resilience; stakeholder theory; sustainable leadership





## 1. Introduction

The COVID-19 pandemic's distinctive nature transformed the business and work environment in ways never witnessed before. The pandemic's duration and ability to simultaneously affect society's micro, meso and macro levels, and spread across geographical and political boundaries distinguish it from all previous global crises [1]. Entrepreneurs are innovative opportunists; therefore, their responses to the pandemic crisis in protecting their enterprises ranged from ceasing operations to 'pass the storm' to opportunity hunting for growth, manifesting their resilience. To explore how entrepreneurs manifested their resilience during the COVID-19 pandemic crisis, EntREsilience [2], a five-country longitudinal case study project [3,4] in China, Malaysia, the Philippines, Thailand and the UK, was launched in 2020 to explore entrepreneurial responses to the crisis.

The literature review for EntREsilience [5] highlighted the gaps in the literature; (1) resilience is a static process, (2) resilience research concentrates on the agents more than the context and (3) resilience is considered the entrepreneur's ability to bounce back 'after' business failure. Therefore, EntREsilience's aim was "to identify effective entrepreneurial firm- and community-level responses and business model practices to support resilient adjustment to the economic adversity triggered by the global COVID-19 crisis" by considering entrepreneurial resilience as "a proactive and dynamic, opportunity-seeking process that seeks to convert the crisis into a source of opportunity". Few studies

have investigated the resilience manifestation by entrepreneurs during the crisis [6,7]. Filling this literature gap, EntREsilience focused on the enabling mechanisms that allow entrepreneurs to adjust to the crisis while it is happening and eventually convert the crisis into a growth opportunity.

Each country team was assigned a different thematic focus to their cases investigating the entrepreneurial responses to the pandemic crisis. The analytical focus of the Thai cases was how community interactions affected entrepreneurial responses to the crisis and moderated their resilience during the crisis. While analysing Thai cases, an added sustainability approach by the entrepreneurs in their response to the pandemic crisis emerged. Thai entrepreneurs applied sustainable leadership strategies [8] for their enterprise's sustainability [9] and as a social mission toward their stakeholders [10]. As sustainable leaders "have a strong propensity towards stakeholder-oriented approaches" [11], this emergent effect led to a reanalysis of the data from the sustainability perspective to investigate if the pandemic crisis response resulted in entrepreneurs adopting any sustainable leadership measures. Data analysis showed that the COVID-19 shock made entrepreneurs recognise the challenges faced by their stakeholders and how enterprise sustainability is dependent upon stakeholder sustainability. The greater awareness engendered sustainability competencies within the entrepreneurs to innovate solutions to their stakeholder challenges, transforming them into sustainable leaders and creating a sustainable business situation for the enterprise.

Through narrating the entrepreneur's journey from March 2020 to February 2022, this article will describe Thai entrepreneur resilience strategies during different phases of the crisis. By extending EntREsilience findings and responding to the identified research gaps [5], the present article will show how entrepreneurial resilience manifestation includes stakeholder crisis-challenge awareness and theorise why Thai entrepreneurs considered stakeholder well-being as one of their strategies to achieve stability and growth. Guided by stakeholder theory [12–14] and applying grounded theory, the article explains how this awareness resulted in sustainable competencies within the entrepreneurs to innovate solutions to those challenges, thus transforming them into sustainable leaders. Replying to the research gaps [5], the present research proposes a transformation model, illustrating 'stakeholder challenges' to the 'sustainability competencies' relationship and how the internal and external contextual factors moderate this relationship. This paper answers the call of Roome, N., and Louche, C. [15] and Liao, Y. [16] that we need more understanding of the processes in a longitudinal study [17] that lead businesses towards sustainability in order to teach future business leaders sustainable business practices.

The following section will overview EntREsilience, its background, research design, findings and limitations; detail the study's theoretical background and present the article's contribution and the research questions. The third section will discuss the research design, cases, and data collection and analysis methods, followed by the findings section which will narrate the entrepreneur's COVID-19 pandemic crisis journey, experiences and responses. Building on the informative story of findings [8], the discussion section will connect the findings to theory to generate an entrepreneurship-to-sustainability conceptual model. The conclusion section will discuss research limitations and implications, and suggest possible future research opportunities and recommendations.

## 2. Research Background, Theoretical Foundation and Research Questions

### 2.1. EntREsilience

The Entrepreneurial Resilience During and After the COVID-19 Crisis Project [2], an inductive longitudinal multiple case study [3,4], was launched in 2020 to understand how entrepreneurs in China, Malaysia, the Philippines, Thailand and the UK manifested resilience in the face of the COVID-19 pandemic crisis. The COVID-19 pandemic crisis provided an opportunity for longitudinal exploration of entrepreneurs' dynamic and proactive measures to protect their enterprises against the pandemic crisis within their respective contexts and use the crisis as an opportunity to grow. EntREsilience focused on

the enabling factors to understand their effect on entrepreneurial resilience. EntREsilience explored the subtle entrepreneurial resilience enabling elements across different institutions and countries, allowing more generalisability [2].

Using the COVID-19 pandemic crisis as the background, EntREsilience was designed to answer two significant research gaps identified by Yuan and Autio [5] in their systematic review of entrepreneurial resilience within the resilience and crisis literature. The resilience literature considers entrepreneurial resilience a static ability displayed after enterprise failure. Secondly, most of the studies are based in a single country, investigating a single case, thus limiting the scope of the results [5]. The review also proposed an integrative framework that emphasises dynamic resilience and opportunity orientation for entrepreneurial resilience in the context of crises, such as the COVID-19 pandemic, and distinguishes between five dimensions of entrepreneurial resilience: agent, context, temporal orientation, enabler and outcome.

In addition to presenting the integrative framework, the review also highlighted seven key findings in the literature on entrepreneurial resilience in times of crisis: (1) the importance of proactive and flexible behaviour for building resilience, (2) the significance of social capital, networks, and relationships for resilience, (3) the role of leadership and strategic decision-making in enhancing resilience, (4) the relevance of organisational culture, values and identity for resilience, (5) the value of learning and knowledge-sharing for resilience, (6) the importance of financial and resource management for resilience and finally (7) the potential benefits of collaboration and cooperation for resilience. The integrative framework and seven key findings from the systematic review formed the basis for analysing the data collected in the EntREsilience project.

EntREsilience defined entrepreneurial resilience as the "ability of the entrepreneur and their business to proactively adapt to, withstand and recover from external adversity; to identify new opportunities during the crisis; and to adapt the business model of the entrepreneurial business such that its ability to pursue new opportunities is strengthened and its robustness against external adversity enhanced". EntREsilience investigated the pandemic crisis effects through its sequence of events, thereby examining it as a process [18]. EntREsilience was designed as an inductive, longitudinal qualitative study analysing entrepreneurs' crisis experiences within their corresponding context before, during and after the crisis. As the entrepreneurs constantly adjusted their responses to the crisis events, an inductive, interpretive approach was the most suitable methodology [19].

EntREsilience data were collected between March 2020 and February 2022 through semi-structured interviews of 45–90 min with "knowledgeable agents" [8] (p. 17), i.e., entrepreneurs. For the choice of these entrepreneurs, see the main report. The case selection was made through theoretical sampling to focus on the focal relationships of interest. Through pilot interviews on company backgrounds, firms with potential resilience enablers of entrepreneurial resilience were selected. China (Wuhan and Shanghai) was used as the exploratory research site. Subsequently, the research was extended to Thailand (Bangkok), the Philippines (Manila), Malaysia (Kuala Lumpur) and the UK. Data collection was performed through three semi-structured qualitative interviews in 2020, 2021 and February 2022. By positioning resilience as a dynamic, proactive and looking-for-opportunities process, EntREsilience examined the "sequence of a crisis" and generated an entrepreneurial resilience process model (Figure 1) using the integrative framework proposed by Yuan and Autio [5].

The first-round interviews were aimed at understanding the businesses as they were before the crisis and the experiences of the entrepreneurs. In the first interview, entrepreneurs talked about how the crisis unfolded, what pressures and challenges it generated, the emotions and feelings of internal and external stakeholders, and how the firm adjusted on the fly and interacted with its community to mitigate the impact of the crisis. The entrepreneurs faced market (demand shock), resource (supply shock) and operational (how to conduct their day-to-day activities) disruptions. Entrepreneurs reacted to these disruptions with ad-hoc and strategic adjustments.

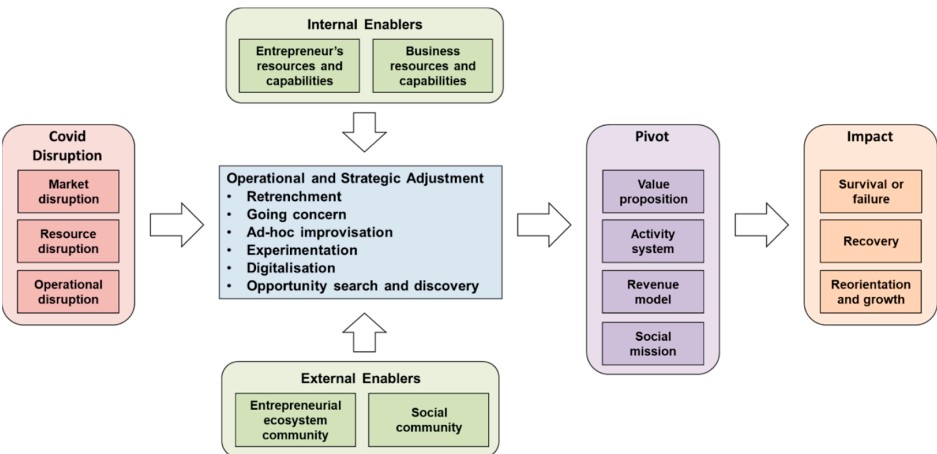

**Figure 1.** The process model of entrepreneurial resilience.

For a few entrepreneurs, adjustment meant altering their business models, refining product/service offerings, value creation and delivery operations. A few entrepreneurs started exploring new markets and customer bases, and some entrepreneurs adjusted their revenue model. For some enterprises, these adjustments were temporary, as they returned to pre-crisis operations after recovering from the disruptions. On the other hand, encouraged by the positively generated results, other entrepreneurs made the adjustments a permanent component of the enterprise. All these adjustments were contextual to the internal and external enablers of the enterprise. Internal enablers included the entrepreneur's human and social capital, capabilities and resources; firm-level capabilities and resources (finance, IT resources); and adaptability. In contrast, the external enablers included the social and ecosystem communities providing the support cushion and learning opportunities (for a detailed description of the model, see EntREsilience [2]).

'Pivots' are the permanent adjustments entrepreneurs adopted to innovate their business models in the face of the COVID-19 pandemic. Dynamic-natured entrepreneurs would choose or discard any adjustment if that adjustment could lead to the stability of the enterprise or its growth. After experimenting with various adjustments, entrepreneurs discovered modifications that worked best for them and made those part of their business model. By the third-round interviews, the focus shifted to post-crisis outcomes to assess the pandemic's impact on entrepreneurs, their businesses and their relationships with their stakeholders. Entrepreneurs discussed the adjustments that worked best for them and those that did not work, the most impactful factors on their 'Covid journey' and their plans for the future. Entrepreneurs also shared how the COVID-19 pandemic impacted their professional growth and maturity.

Data collected from the interviews, observations, field notes and any online information were triangulated [3,20] to compile a detailed narrative for each case. Data collected from the three interviews had a timeline: (1) firm operations before the pandemic crisis; (2) experiences when the crisis started and progressed; (3) lessons learnt and anticipations; and finally, (4) post-outbreak response strategies and reflections. Each research site analysed data across cases through open coding using Atlas.ai, and developed resilience concepts using Corley and Gioia's practice [21] and Gioia, Corley and Hamilton's [22] guidelines to compare and find similarities, differences and potential patterns across cases forming a 'data structure' to summarise the emerging concepts and themes. A grounded theoretical model was followed to illustrate the entrepreneurial crisis-response process and their adjustment strategies. The next step was to strengthen the validity and clarity of the emergent constructs by consulting the crisis and resilience research literature. The final constructs were then used for the process model (Figure 1) development to show entrepreneurial firms' learning and adjustment process throughout the pandemic crisis within their contextual factors.

After analysing the first-round interviews, five pairs of 'polar-outcome' or contrasting responses to the COVID-19 crisis were observed. (1) Businesses actively changing their business model (proactive) vs. passive businesses 'waiting' the crisis out (reactive). (2) Businesses that rapidly grew (grow) vs. enterprises that suffered (suffer). (3) Businesses that relied closely on their community support and advice (community-dependent) vs. businesses that shunned community interactions (not community-dependent). (4) Businesses that successfully leveraged digital technologies (digitally enhanced) vs. businesses that could not or would not increase digital technology usage (not digitally enhanced). (5) Businesses changing their business model (change business model) vs. businesses only changing their product and service offerings (change product/service). For the second and third interviews, each research site was assigned one 'polar pair' to focus on; the Thai team was to focus on how community interactions affected entrepreneurial responses to the crisis and moderated their resilience during the crisis.

The process model does not explain all these subtle details (Figure 1). Similarly to Duchek [23], the EntREsilience process model demonstrated how entrepreneurs manifested resilience in the face of crisis without explaining why entrepreneurs made certain contextual adjustments and how the enablers affected their adjustment response. The enablers include organisational stakeholders (workers, suppliers, customers and social connections). Santoro [24] emphasised comprehending the nature and calibre of social ties [23], how entrepreneurs connect with their stakeholders and how these different connections affect their resilience towards entrepreneurial success. The EntREsilience model does not elucidate how the pandemic crisis changed the quality and nature of entrepreneurial interactions with their stakeholder to maintain critical business functioning [25]. It does not differentiate between the social and ecosystem stakeholder (Figure 1) involvement in explicating their influence on the entrepreneur's response to the pandemic crisis [26]. As the process model shows linear phenomena, it does not clarify how the entrepreneurial adjustments affected the stakeholders, neither does it illustrate the contribution of the entrepreneurs in the lives of their stakeholders during the pandemic crisis.

Shared location, identity, fate, interest and practice form communities that, through inter-community interactions, can support entrepreneurial activities [27]. EntREsilience data showed entrepreneurs remained active members of their communities during the pandemic crisis, creating value from community interactions [28]. The EntREsilience Thai team investigated two community interactions. The social community interactions, i.e., interactions with family and friends, offered the entrepreneurs psychological and moral support, thus augmenting their resilience. Secondly, ecosystem community interactions, i.e., with workers and co-workers, suppliers and customers or members of the same craft or industry [27], provided entrepreneurs with the technical tools and knowledge that let them run their businesses smoothly. Analysis of the community interactions quickly revealed an emerging phenomenon. As the entrepreneurs were 'living the crisis' themselves, their community interactions provided direct feedback on how the crisis disrupted the lives of their suppliers, customers, workers and other enterprise stakeholders. This realisation engendered an altruistic spirit within them, whereby entrepreneurs initiated and adopted measures that, while supporting their enterprise, were aimed at the sustainability of their stakeholders. This new realisation and the limitations of the process model mentioned earlier demand an additional investigation to understand the nuanced effects of community interactions on entrepreneurial resilience.

*2.2. Theoretical Background*

EntREsilience described resilience as the capacity of the entrepreneur and their business to proactively adjust to adversities, recover and identify new prospects in the crisis. Organisational resilience is a wide range of capabilities, capacities, characteristics, outcomes, processes, behaviours, strategies and approaches [29]. The literature has constructed resilience at different levels, emphasising that organisational resilience is achieved at collective levels through employees and teams and their collective actions leading to a shared

vision. According to Duchek [23], the presence of a shared vision is integral to a resilient response, facilitating the implementation of effective solutions to address challenges presented by an adversary. Furthermore, Torres [30] suggests that stakeholder connections serve as a valuable long-term resilience asset by providing entrepreneurs with a network of support and assistance.

Stakeholder theory considers any business as a set of relationships among buyers, sellers, workers, investors, communities and executives who work together to create mutual value [13,21]. Stakeholders are "those groups and individuals who can affect or be affected" by business functions to create value [12] (p. 9). The relationships between a business and its stakeholder are the main focus of Stakeholder theory [12,31]. Stakeholder engagement refers to the active involvement of stakeholders in an organization's decision-making and management processes. This involvement leads to greater awareness among leaders of the concerns and problems faced by stakeholders, which in turn leads to greater challenge awareness, more interaction and feedback and adjustments for better decision-making and more effective actions [32]. Stakeholder engagement is viewed as a critical enabler of collaboration achieved through sharing, cooperation, networking and partnership [33]. In fact, Tolkamp et al. [34] suggest that stakeholder engagement is essential to a business's success, particularly in generating social solutions, fostering a collaborative perspective and facilitating elevated levels of economic, social and environmental innovation that lead to the creation of inclusive value.

Stakeholder problem awareness, a prerequisite for effective stakeholder engagement, requires understanding the nature of stakeholder interactions [21]. Stakeholder problem awareness is critical for business leaders who wish to promote the well-being of their stakeholders and adopt a sustainable leadership style [35] to develop sustainable businesses [28,36]. When the ideals of sustainable business models are incorporated into interactions with stakeholders, business practices move towards sustainability [37]. Sustainable businesses capture economic value for the business without endangering natural, communal and financial resources that extend beyond a business's borders [38] (p. 6). Sustainable business models prioritise stakeholder relationships and shift the focus from solely creating value for customers, suppliers or business partners to creating value collaboratively with stakeholders [14] by aligning the stakeholders' interests and demands [39]. Sustainable business models address social and environmental issues by leveraging high levels of economic, social and environmental innovation, creating inclusive value [34,37] while avoiding the depletion of natural, social and economic capital [38] (p. 6).

Avery and Bergsteiner proposed a "Honeybee" leadership model with twenty sustainable leadership sets of behaviours and practices and three drivers aimed at creating lasting value for all stakeholders, including society, the environment and future generations [8,40]. The three key performance drivers, innovation, staff engagement and quality, are critical to organizational performance. The Honeybee leadership model extends beyond the traditional triple bottom line to achieve outcomes that enhance brands, customer satisfaction, and long and short-term financial viability, while providing long-term value for all stakeholders [41]. Six schools of thought within sustainable leadership scholarship relate to how leadership contributes to organizational sustainability and long-term value for the stakeholders. The six schools of thought are sustainable leadership, leadership for corporate sustainability, managerial leadership, responsible leadership, ethical and transformational leadership and leadership for sustainable change [42]. Firms implementing sustainable leadership principles are more likely to achieve sustainable performance outcomes, long-term resilience and stakeholder satisfaction [41].

Sustainable businesses prioritise long-term outcomes over short-term gains for sustainability and resilience. Leaders who adopt a long-term orientation tend to emphasise future-oriented actions and outcomes rather than short-term goals [40]. A long-term orientation requires diverse sustainability leadership competencies, including strategic and systems thinking, and foresight competence, to ensure that organizations can antici-

pate and respond to the complex and dynamic challenges associated with sustainability and resilience.

*2.3. Research Questions*

The EntREsilience Thai team were intrigued when during the second and the third interview, some of the entrepreneurs, guided by awareness of their stakeholder problems, were actively initiating measures towards their stakeholder sustainability. Sustainable leaders have a strong tendency to take the interests of all stakeholders into account rather than solely focusing on financial gains [11]. The EntREsilience Thai team wanted to reanalyse the data within the theoretical framework of stakeholder well-being and sustainable leadership.

The literature recognises that sustainable leadership behaviour has a transformative character, necessitating the study of its antecedent variables, moderators and mediators [15,16]. Similarly, the business model for sustainability literature has called for a greater understanding of stakeholder networks for mutual value creation [14] and learning from the surviving businesses [43]. By seeking to answer these calls, the present research aims to extend the entrepreneur resilience-manifestation process model (Figure 1) to understand the underlying causes of what had happened and explore (1) if and how the disruption of the COVID-19 crisis resulted in an altruistic spirit within entrepreneurs, fostering greater stakeholder-wellbeing awareness; (2) how this awareness generated sustainability competencies within entrepreneurs and; (3) what contextual factors moderated the awareness to sustainability competencies development relationship?

## 3. Methodology

Following the EntREsilience methodology, an inductive, qualitative research approach to investigate little-known phenomena [44] was applied in the present study. No new data were collected. The EntREsilience data of selected Thai entrepreneurs were reanalysed through an interpretive paradigm [19]. The EntREsilience data were collected with different thematic foci across different research sites. Thai cases were investigated with a focus on community interactions. The data collected from a few Thai cases indicated a community-interaction effect on the stakeholder challenge awareness, which prompted the present study. Therefore, it was pragmatic to exclude non-Thai cases and Thai firms with low or no community interactions to keep the focus on the central relationships of our interest.

*3.1. Respondents*

Table 1 summarises the selected Thai entrepreneurs whose interview data were used for this article. This representativeness [45] of the purposive sample of entrepreneurs provided an appropriate variation on the sustainability dimensions facilitating generalisability building [20,46,47]. As this research reuses the EntREsilience data, no additional interviews were conducted. Familiarity with the data when analysing them for EntREsilience aided in case selection.

**Table 1.** Summary of the four cases.

| Company | Industry | Respondent | Business Model |
|---------|----------|------------|----------------|
| EJ | Jewellery manufacturing | Founder and Director | Providing one window manufacturing service for jewellery retailers. |
| EG | Garments and fabric | Co-Founder and CEO | Creating value through the circular economy by marketing the leftover fabric of garment manufacturers. |
| ER | Retail | Family Entrepreneur and Director | Transforming traditional Thai retail into a hybrid industry via digital marketing. Developing Silver Entrepreneurs and encouraging local producers through supply chain services. |
| ET | Tourism | Co-Founder and Director | Customised tour packages for tourists looking for a different experience. |

### 3.2. Data Analysis

As an exploratory follow-up study to EntREsilience, this study repeated the EntREsilience methodology to code and analyse data. Open coding of the transcripts was performed using Atlas.ai. Gioia, Corley, and Hamilton's [22] systematic coding methodology of theory development was followed for sense-making [21], how the COVID-19 pandemic made entrepreneurs aware of stakeholder challenges and adopted sustainable leadership traits. Open coding the rich qualitative data allowed us to reduce the large volume of materials and generate an initial set of first-order codes, which were informant-centric and contextually embedded.

Data analysis started with the re-coding of the interviews. In their interviews, entrepreneurs described the events that happened from March 2020 to the third interview in February 2022 and what measures they took to survive the pandemic crisis, recover from it and create a sustainable growth path. Therefore, the interviews narrate the interviewee's journey through the pandemic crisis. The analysis of this journey generated 55 codes that were grouped into ten first-order themes. Then, the first-order codes were categorised into second-order themes through axial coding. The three-part second-order themes extracted from the first-order codes are groups of the events and measure code categorised on a higher level of abstraction, allowing a structured presentation of data and rigorous inductive analysis [8,15].

The second-order themes are the entrepreneur's 'Covid Journey', developing or 'Building Sustainability Competencies', and the 'Moderators' (Figure 2). The structured data representation also enabled iterations between the findings and the extant literature, comparing the two to articulate the constructs from the findings to answer the research question. This article's 'Findings' section will elucidate the second order and aggregate the theme with interview quotes. The second-order themes merged to form the third-order aggregate 'Sustainability Mission' (Figure 2). This aggregate 'Sustainability Mission' explains how the pandemic crisis created a situation that refined the entrepreneur's interaction with other stakeholders towards sustainability. The Thai team analysed interview data by examining community dynamics as a moderator to the entrepreneur's resilience manifestation. The multiple-case-study approach helped in "identifying and refining constructs and their relationships" [10] (p. 373). Community factor analysis revealed the nuanced way entrepreneurial resilience is shaped when the entrepreneurs are more conscious of their enterprise's internal and external stakeholders, considering stakeholder sustainability as integral to their success.

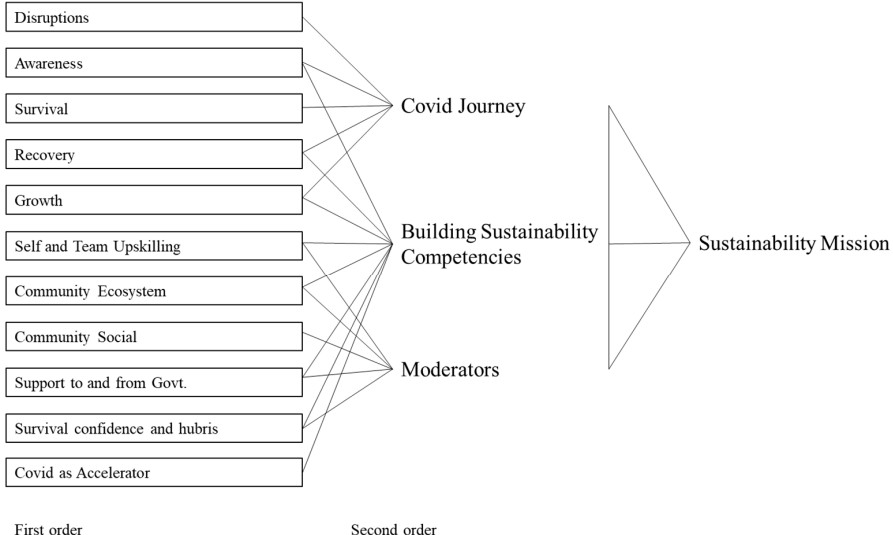

**Figure 2.** Data structure showing the coding tree and aggregate dimensions.

The findings section will explain the coding tree by narrating the entrepreneur's COVID-19 crisis journey with related quotes from the interviews to show how they developed sustainability competencies to solve their stakeholder's challenges.

## 4. Findings

This section will describe the findings from the inductive exploration of interview data and the resultant data structure. As the interviews were entrepreneurial experience narratives, the findings are presented temporally. This section is organised to answer how the crisis conditions fostered sustainability competencies in the entrepreneurs towards their stakeholders, making them sustainable leaders, and the possible role of moderators.

The Thai government, in response to the first outbreak of COVID-19, announced emergency measures on the 26 March 2020, followed by containment measures like lockdown and banning incoming international flights leading to economic disruption, difficulty in business operations and loss of work [48].

> **EJ**: "in April 2020 [ . . . ] it was very difficult because I had to run my business as well. I have to pay salaries, but the work was not there."

> **ER**: "The vendor cannot come and sell, the people were locked down".

> **ET**: "My business stopped. They were closing the country to foreign tourists indefinitely. We [do] not have customers".

Distressed and confused entrepreneurs started feeling that they were losing control as no one had ever experienced this kind of global-level extended disaster before.

> **EG**: "In May 2020 [ . . . ] anxiety, anticipating everything, no peace of mind. I don't know what to do. I'm out of control of everything".

The uncertainty added to pandemic crisis distress and confusion. In the first few weeks, no one had a clear answer about the nature of the crisis and how it will be resolved or end.

> **EJ**: "It would just eat up your whole full year. People are thinking that its 3 or 4 months or 7".

> **ET**: "I thought it will be a matter of just a few months and then everything will resume the way it was."

Nevertheless, soon, entrepreneurs realised and accepted that the pandemic was there for an extended period and would have a profound impact on their businesses and that they had to start strategising accordingly.

> **ER:** "COVID would be with us. Not just only this year, but perhaps the next or two years or until 2024".

> **ET**: "I began to feel worried for the sustainability of my business."

This also meant that the entrepreneurs had to start implementing measures to protect their businesses. Therefore, they began applying survival measures, starting with themselves. Entrepreneurs are inherently flexible and hands-on. The first step for the resilient entrepreneurs of smaller enterprises with no external financial resources was to ensure they have vital funds for survival by exploring options outside their enterprise.

> **EG**: "I also took on another part time job to make sure that I have some cash".

> **ET**: "Main challenge is surviving by doing multiple unstable jobs."

The business operation refinements further demonstrated the entrepreneurial flexibility for the enterprise's survival.

> **EG**: "First few months were like, OK, let's close the operation. Eat from savings and let's see what happens".

> **ET**: "Survive by furloughing our five employees and by closing down our physical shop."

From the beginning, the entrepreneurs adopted survival measures, revealing an awareness of the challenges faced by the enterprise's stakeholders [49]. Stakeholders affect and are affected by the value creation activities of a firm [12,13]. The relationships between the stakeholders and their joint effort are crucial for establishing and solidifying an enterprise's value creation network [14,50]. Both internal stakeholders (owners and employees) and external stakeholders (customers, suppliers, governments, ecosystems, local communities and environment) are essential for the enterprise's success, but it is primarily the internal stakeholders that enable a firm to fulfil stakeholder demands [51,52]. Entrepreneurs were attentive to the predicaments faced by the stakeholders, and they knew that the path to enterprise sustainability had to go through solving stakeholder challenges.

> **ER**: "So all people cannot work [ . . . ] old and young people [ . . . ] no hiring's from the entrepreneur. [ . . . ] Both generations go back [to the village]".

> **EJ:** "After learning that all this company is shutting down, few workers, they come and ask me for that [ . . . ] How should we do? Because If the company cuts the salary they don't have enough money to pay [ . . . ] everybody has bills".

However, the initial survival strategy involved laying off the staff. The entrepreneurs were fully aware of how vital stakeholder relationships are; therefore, even for such a complicated step, they showed prudence in being mindful of inter-stakeholder relationships.

> **ET:** "When the hard moments of cutting salaries or sending our staff back home came everybody understood and everybody agreed to keep relations the best we could".

> **EJ**: "In April we were down from 28 to 10 workers. The permanent staff was only retained [ . . . ] It would be much better to keep the most required 10 people and increase [ . . . ] surviving from three to maybe 6 or 7 months."

Awareness of the stakeholder difficulties and adopting measures that ensured stakeholder welfare disclosed the entrepreneur's realisation that the COVID-19 pandemic is a protracted crisis [49]. Therefore, they required a strategy that could sustainably counter the disruption by strengthening the enterprise's relations with its stakeholders [53,54]. The strategy necessitated the entrepreneurs to develop sustainability competencies and adopt practices to evolve into sustainable leaders for enhanced business resilience and performance [8,41]. After surviving the initial stage of the pandemic crisis, the entrepreneurs felt enough confidence to look for opportunities and refine the survival plan in a sustainable recovery-to-growth business strategy.

> **EG**: "I feel more confident that I should now be offensive on this business. Looking very thoroughly on what we have and how can we make money out of whatever? Forever. Seriously."

Entrepreneurs, as opportunistic innovators, knew that their recovery-to-growth strategy could only be effective against the pandemic crisis if it were sustainable with a long-term perspective [55]. Business sustainability requires trusting and innovating teams with a shared vision [56] and a strategy that meets "the needs of a firm's direct and indirect stakeholders" [57] (p. 13), stimulating business resilience. As per their recovery-to-growth strategy, the entrepreneurs refined their business operations displaying their sustainability competencies toward social sustainability [58]. The modified business operations focused on offering new services based on stakeholder relationships, revising organisational structure to develop people and processes for team-centred efficiency and bolstering external stakeholder relationships through joint-venture.

> **ET**: "Launched free tours around Bangkok [...] only at the end of the tour customers can leave tips for the guide [...] a cost-effective solution able to continue to spread our company tours and which could help with a little income our guides."

> **ER**: "We tend to have [ . . . ] Faster decisions, [so] if a department come to their SWAT team and shoots out specific project details, and we do this together for like a month and we finish."

> **EG**: "We work very well with some of our collaborations, sometimes we ask them to become the designer of our big deals. We give them a share."

The entrepreneurs knew that "value should be created both with and for different stakeholders" [14] (p. 5) for a sustainable recovery-to-growth; therefore, all the implemented plans were centred around stakeholder development, reaffirming the entrepreneur's commitment toward stakeholder relationship development. Each business modification, aimed at mitigating stakeholder challenges, covered one or more sustainable leadership practices [8], refining the internal stakeholder relationships by providing a positive feedback loop for the entrepreneurs. In the second interview, a couple of entrepreneurs were already talking about trusting, cohesive teams and better streamlined, efficient work, which they credited to the mentioned positive feedback that added to their confidence and ensured the enterprise's trajectory towards sustainability and resultant growth [9].

> **EJ**: "So, I assured my workers and I told them that I will not slash their money, I will try as much as possible [...] made me gain their trust as well for the future because even as a small company, but still the boss is keeping the interest of the workers more than interest of himself."

> **EJ**: "My people, they support me. I try to support them."

The recovery and recovery-to-growth strategies and resultant modified business operations demanded that entrepreneurs advance self and team skill development towards a more sustainable business offering better service and aiming for customer-based expansion. By reading, listening to experts and taking online courses, entrepreneurs improved their knowledge and skills. For team skill development and enhanced team cohesion, entrepreneurs pushed themselves to be better leaders, guiding and coaching their teams through discussion and training to be more efficient and productive.

> **ET**: "my husband [co-founder] is determined to take a proper course to deepen his knowledge in digital marketing".

> **EG**: "I started to [go] back to exercise and start reading selective books and listen to the podcast quite often."

> **ER**: "Be a good coach to my team".

> **EJ**: "To have more skilled people [so we make] minimum mistakes. So we can give to customers who do not opt for some other place".

Stakeholder development, stakeholder relationship development and team skill development lead to social sustainability [58] and better social capital [59]. Social capital is the network that facilitates resource offerings for the enterprise from formal and informal sources. The trust and reciprocity of the sources develop social capital [60]. As part of the sustainability mission taken by the entrepreneurs directed toward social capital growth for enterprise sustainability, entrepreneurs started fostering and advancing links with the business community and public institutions to help and share ideas with them [30].

> **ER**: "Helping the communities and helping the silver population. [We] have a course to teach about digital literacy, financial literacy, how to take care of themselves, how to have health awareness."

> **EG**: "I share the circular mindset [ . . . ] to scale the mindset and convince people [within incubators] that with this mindset, you can make some money [from the circular economy]. I'm one of the five startups representing Thailand for the Innovation Promotion, for the Foreign Ministry and National Innovation Agency NIA."

These intentional interactions resulted partly from the altruistic values and expected synergies with other entrepreneurs or public institutions. The entrepreneurs got a positive response to their efforts with better external stakeholder relationships. External stakeholders replied with deeper trust, firmer support, positive EWOM and even assistance from international development agencies and Thai public organisations. The external stakeholder's positive feedback added to the business's competitive advantage [10], furthering its sustainability.

> **EG**: "We have satisfied our international customers, our international customer came back and trusted us in delivering their products".

> **EG**: "The incubators [ . . . ] give me some visibility [ . . . ] put me on the list of potential companies to invest in to be on the UNDP website".

> **ER**: "Government asks [us] to participate in [ . . . ] exhibitions for 15 cities in Thailand to show the application of our platform that can help [the rural population] to turn their home to be their retail store. This is really good publicity from the government because it's showing that my business model is working."

All Thai entrepreneurs defined their social community, mentors and family members as helpful and motivational sources that energised entrepreneur resilience through their support.

> **EJ**: "I got support from my family [ . . . ] And that support sometimes gives you the strength to carry on for a little bit longer."

> **EG**: "One thing significant. Last year, [ . . . ] I start to have a mentor [ . . . ] works out amazing, amazingly well. Because mentor to me is like a Coach."

The last interviews were conducted in February 2022. Although Thailand was reporting a high number of cases [3] at that time, after multiple vaccination cycles, it became apparent that the pandemic crisis had become manageable [Bangkok Post, 27 January 2022]. Each entrepreneur was asked retrospectively about their two-year experience and how they would summarise the effect the crisis had on them. Not surprisingly, all of them agreed that the COVID-19 pandemic was an accelerator [61] to their professional development, contributing to their resilience and improving their business leadership qualities.

> **EJ**: "I have actually got maturity of this two years much more than maybe I would have been in five-six years".

> **EG**: "I think COVID has helped me as a business founder to be more rounded and, more grounded".

> **ER**: "[pandemic crisis] is a good thing for the leadership, for a leader to have a change. Incubator program, I have this kind of concept for the last 4 years. We can do this incubator in just two months [because of the pandemic crisis]."

> **ET**: "I believe this whole situation has taught me quite a lot."

## 5. Discussion

The COVID-19 pandemic was a once-in-a-century crisis. It changed how the world operates and how we perceive it. When the pandemic crisis hit the entrepreneurs, they had to innovate tactics to survive and convert the crisis into a growth prospect, exhibiting their resilience. The EntREsilience [2] resilience process model (Figure 1) explains how entrepreneurs manifest their resilience, and its principal mechanisms and enablers. This research builds on the findings of the EntREsilience project, answering [14,16,62] to exhibit how one of the enablers, the entrepreneur's community interactions, contributed to the resilient entrepreneurial pandemic response [9,63]. The pandemic crisis engineered a response centred around the stakeholder relationship and well-being [64], propelling the entrepreneurs to develop and demonstrate sustainability competencies [41] and become sustainable leaders [42] as they resiliently converted the pandemic crisis into a growth opportunity for their enterprise sustainability [9].

The lockdown announced by the Thai government in April 2020 to counter the COVID-19 pandemic outbreak disrupted all normal life processes, including how businesses operate. In the first few uncertain months of the pandemic crisis, the Thai entrepreneurs felt confused and anxious, unable to manage the exogenous disruptive events. Lack of information and the absence of any indication of how long the crisis will be and how it will end amplified feelings of losing control. The entrepreneurs needed to survive the disruption and keep their enterprise functional, even if it meant minimum functionality. When the Thai entrepreneurs realised that the pandemic crisis would be a prolonged disruption affecting everyone, they knew they had to develop strategies and shape their response step by step, starting with surviving the initial wave of disruption. Survival meant financial stabilisation and keeping the enterprise in working condition. How they shaped their survival revealed how the entrepreneurs started developing sustainable competencies [41,55], transforming them into sustainable leaders [56].

To be financially stable with minimum cash flow, some entrepreneurs even took part-time jobs, affirming their flexibility and hands-on resilience. A minimum-functioning enterprise necessitated cutting expenses by modifying operations, including laying off workers. The COVID-19 pandemic crisis is an extended global crisis affecting everyone; no one has escaped the disruptions caused by the crisis. This crisis characteristic has enabled entrepreneurs to be mindful of their stakeholders' hardships. When they had to furlough workers, by valuing their relationship with their workers [65], the entrepreneurs displayed concern and awareness of the challenges their enterprise stakeholders faced and would face in the coming days. The way they laid off their workers in an amicable manner set the course for all of their future crisis-response steps in the coming months, stakeholder-centric with a long-term vision [8,41].

Surviving the initial wave of disruption and stabilising their enterprise at a minimum operating level boosted the entrepreneur's confidence and resilience. They started strategising a recovery plan that could lead to a sustainable enterprise with financial stability and resilience [9]. Entrepreneurs were clear that a sustainable enterprise requires furthering stakeholder relationships with increased social capital [60], maximising value for a wide range of stakeholders, specifically through knowledge sharing, so that its societal effects are felt beyond its organisational boundaries, all with a long-term perspective [38,62,63,66]. The internal stakeholders (in this case, the employees) enable a firm to fulfil all stakeholder demands [52]; therefore, to create value "with and for" [14] (p. 5) their employees and strengthen their relationship with them, entrepreneurs took several initiatives. By acquiring additional skills through various resources and becoming better coaches to their team members, the entrepreneurs fashioned an innovative, cohesive and trusting team with a shared vision and better skills, transforming the stakeholder relationship nature [67] to deliver stakeholder satisfaction for sustainable recovery [41,68].

A prosperous stakeholder is central to an enterprise's value-creation activities in a connected world. As the value creation network of an enterprise is highly dependent on inter-stakeholder relationships, entrepreneurs were alert to the predicaments faced by all of their stakeholders. Whether it was an issue of customers having difficulty in buyer interaction or their vendor's inability to operate their business, the entrepreneurs were attentive to all these developments [64]. Enabled by a trusting, cohesive team, they started working on extending their social capital network [59] and growing external stakeholder interactions, securing improved social and economic value [69] and transforming their relationship nature [67]. These interactions were with social and business community members, offline and online, on private, governmental or international forums. The results were expanded links within the social and business community, firmer ties with public institutions to understand their challenges and better idea sharing [30]. The social community interaction with friends and family boosted the entrepreneur's morale. Mentor interactions acted as a 'perspective catalyst'; mentors enhanced the entrepreneurs' ability to empathise and connect with their stakeholders by discussing novel ideas.

Even though the pandemic crisis has disrupted businesses, it created many new opportunities that innovative and repositioned business processes could benefit from. The entrepreneurs extended their recovery plan to a growth plan to gain from as many new opportunities as possible. Driven by monetary motivation and societal sustainability concerns, the above interactions enabled knowledge sharing on how to survive the crisis and change it into a growth opportunity. Along with better external stakeholder relationships, the external stakeholder interactions resulted in positive feedback, increasing the business's competitive advantage [10]. Interaction with public institutions and organisations led to better financing options for one entrepreneur, whereas another's business model was validated by governmental endorsement, promoting their brand.

The COVID-19 pandemic journey started as a crisis which disrupted the entrepreneur's business operations, creating confusion, anxiety and loss of control. Entrepreneurs, guided by their values and beliefs, recognised that their entrepreneurial sustainability is connected to the sustainability of their stakeholders. By increasing their interactions and deepening their connections, the entrepreneurs developed a greater awareness of their stakeholder's challenges. The greater awareness engendered sustainability competencies within the entrepreneurs, enabling them to innovate sustainable solutions to their stakeholder challenges. The sustainability missions toward internal and external stakeholders generated feedback to the entrepreneur's sustainability mission, positively regulating it (Figure 3).

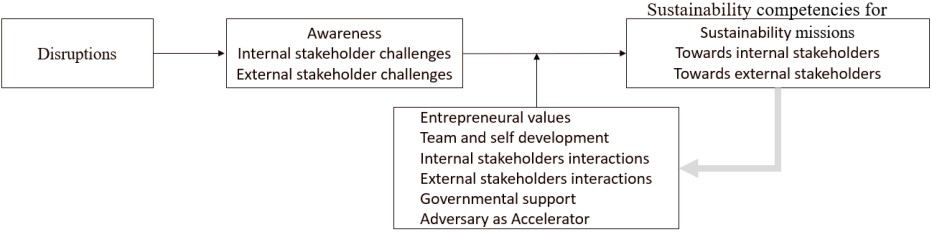

**Figure 3.** Transformation of entrepreneurs into sustainable leaders: how the COVID-19 crisis transforms entrepreneurs into sustainable leaders.

Figure 3 shows how the entrepreneurs, propelled by the stakeholder challenges, transformed into sustainable leaders for their enterprise, teams, and external stakeholders. The model agrees with Duchek [23] that social resources positively influence resilience and aligns with the model proposed by Sontoro [24], as the stakeholder interaction moderates the entrepreneurial crisis response in both models. The transformation model adds to the process model (Figure 1) to explain the role of stakeholder challenge awareness in entrepreneurial resilience. The transformation model depicts how stakeholder challenge awareness resulted in the entrepreneurs adopting sustainability missions directed toward the stakeholder's well-being. This awareness prompted sustainability competency development in the entrepreneurs, encouraging sustainable measure adoption for creating a sustainable business situation for the enterprise. This transformation model adds to the process model (Figure 1) by showing how stakeholder challenge awareness facilitated business operational and strategic modifications with the addition of sustainability measures. The transformation model lists moderators affecting sustainability competency development as modifiers and enablers.

Quite a few of the sustainability measures adopted by entrepreneurs are similar to those in the sustainable leadership pyramid [8]. Sustainable leadership pyramid research [8,65], based on the 2008 credit crunch crisis, explored how corporations adopted a stakeholder approach for business sustainability. Similarly to the COVID-19 pandemic crisis, the businesses and their stakeholders were 'living the crisis'. Corporations such as BMW [65] realised stakeholder challenges and refined their usual business practices, adopting sustainability competencies and practices to evolve into sustainable enterprises. Entrepreneurs in the present study displayed a similar course of action as their pandemic crisis response, however, with few nuanced differences.

Sustainability demands a balance between environmental care, social well-being and economic growth, and one of the foundational sustainable leadership practices is environmental responsibility [8,62,70–72]. In the past three centuries, our economic system has developed by exploiting 'commons' where business revenue, profitability and shareholder dividends are never calculated by accounting for the natural resource depletion cost [73]. A big corporation such as BMW has environmental responsibility as a core to their operations [65] as they can afford to do so; however, due to financial performance requirements, SMEs seldom consider the natural environment as their primary stakeholder and thus, its care is not considered much in the decisions made [74–76]. The present research found similar tension between sustainability and economic missions [10], as even when the entrepreneurs exhibited an attitude of social responsibility, they never mentioned or discussed environmental responsibility in their interviews. In their quest to survive, recover and grow from the COVID-19 pandemic crisis, the entrepreneurs had environmental sustainability as their lowest priority [77].

Entrepreneurial sustainability competencies development and practising sustainable leadership traits for social sustainability initiatives were initiated for the economic sustainability of the enterprise. The results of the present research disagree partially with Suriyankietkaew [55], as the entrepreneurs' initiatives to ensure internal and external stakeholder well-being were not the goals of the entrepreneurs. The entrepreneurs considered better stakeholder relationships and stakeholder well-being as one of their strategies to achieve financial stability and economic growth. None of the entrepreneurs talked about nature as their business stakeholder [74,78], as SMEs do not consider environmental consciousness to translate into financial rewards [79,80]. Therefore, can the entrepreneurs be called sustainable leaders when they only consider social sustainability a strategy for economic sustainability and do not have ecological sustainability as part of their business practices? The answer to this question lies in our economic system and growth fundamentals [73], which demand further theoretical and empirical research. Once we answer such questions, we can motivate entrepreneurs to consider nature as their stakeholder and give ecological and social sustainability similar priority to their enterprise's economic sustainability.

## 6. Conclusions

EntREsilience, a five-country longitudinal study, was launched in 2020 to understand how entrepreneurs manifested resilience in response to the COVID-19 pandemic crisis by proposing a resilience process model (Figure 1). Present research added to the process model by analysing the strategies applied by the entrepreneurs in their response to the pandemic crisis. This study is consists of exploratory case study research. The research data were collected by interviewing four Thai entrepreneurs in December 2020, May 2021 and February 2022. In the interviews, the entrepreneurs described events from March 2020 to February 2022 and how they responded to the different phases of the crisis. The research focused on exploring the sustainability measures adopted by the entrepreneurs as their response to the crisis. The inductive results of this longitudinal study showed how the awareness of their stakeholder challenges shaped the entrepreneurial response. Realising the importance of stakeholder well-being to the sustainability of their enterprise motivated the entrepreneurs to develop sustainability competencies towards their stakeholder's challenges, applying sustainable leadership traits. Moderators affecting entrepreneurial responses were also examined.

This research has contextual limits. The COVID-19 pandemic was an extended crisis affecting everyone, irrespective of socioeconomic status. Empathising with and understanding the hardship of stakeholders was easier as everyone was 'living the crisis'. Will the entrepreneurs respond with similar sustainability measures to a crisis if it only affects a part of the society? If the crisis is local to the enterprise only (for example, market loss), what sustainable measure will constitute the resilience manifestation of the entrepreneurs? Another limitation is the lack of contrast cases in this study. All the Thai entrepreneurs interviewed

for EntREsilience were able to survive the pandemic crisis. A contrast case, where the entrepreneurs had to close their business and stop being an entrepreneur (e.g., by taking employment), would have revealed subtler details to the entrepreneurial resilience literature. The purposive sampling only analysed entrepreneurs involved with their communities, whereas random sampling would have provided different entrepreneurs' response details. Thailand is a collective society where community interaction is part of daily life, but the same cannot be said about individualistic societies. Therefore, can the same phenomena be observed in individualistic societies where community interaction generates sustainability competencies within business leaders?

This research extends the EntREsilience process model in the context of sustainability and stakeholder relationships. Awareness of their stakeholder challenges refined entrepreneurial pandemic crisis response as they adopted sustainability traits, making them sustainable leaders. The research also showed that working with external stakeholders enhanced enterprise sustainability by providing entrepreneurs with new growth opportunities. The research reaffirmed the centrality of economic sustainability to entrepreneurs, where social and ecological sustainability were considered only by their contribution to economic sustainability.

### 6.1. Theoretical Implications and Future Research

The research inductively explored an area of knowledge where entrepreneurial crisis resilience, stakeholder theory and sustainability measures merge. This study explored how being aware of stakeholder challenges can endanger sustainability competencies within entrepreneurs directed toward the stakeholder's well-being, producing social sustainability and enabling enterprise economic sustainability. Future research can deductively investigate these factors to explicate each construct's relative effect in generating sustainability measures.

Specific conditions (for example, crisis duration) affected the entrepreneurial resilience with subtle sustainability traits, as reported by EntREsilience and the present research. By exploring resilience manifestation, facing adversaries in various crises might highlight the specific conditions when entrepreneurial crisis response will satisfy all three sustainability conditions to make them sustainable leaders. One interesting observation in the EntREsilience project was the birth of 'Covid-babies', a business born out of the COVID-19 crisis. Exploring their sustainability measure will add to understanding the nuanced entrepreneurial attitude toward social and ecological sustainability. This research underlined the role of social community and mentorship in moderating an entrepreneur's resilience. The function of social community and especially the role of mentors in shaping entrepreneurs to be sustainable leaders may be a good area for future research.

### 6.2. Managerial Implications and Recommendations

This research showed the importance of social sustainability for entrepreneurs in achieving economic stabilisation and sustainability. Positioning stakeholder challenge awareness as an essential part of leadership training will ensure business sustainability. By making social sustainability an important part of organisational culture for achieving economic growth and sustainability, the managers and business leaders may use the inertia to generate societal sustainability. The spill-over effect of social sustainability can also be directed toward ecological sustainability for a sustainable future.

This research validated a community's role and influence [81,82] in entrepreneurial resilience. Social community interaction, specifically mentorship, boosted the entrepreneur's resolve and belief in themselves and motivated them to face the pandemic crisis. Promoting mentorship within entrepreneurial communities can be an efficient source for entrepreneurs to discuss their personal and business problems, exchange innovative ideas and learn from the mentor experience. Public institutions should coordinate platforms where entrepreneurs can meet potential mentors. In the collective Asian societies, 'angel

mentorship' can be effective and may lead to a two-way exchange of knowledge sharing, as illustrated by the ER case.

Business communities for social exchange promotion can be an effective synergy platform. The digital meeting apps saw a massive rise in subscriptions when the pandemic-induced restrictions were imposed, proving their effectiveness. Exploiting their potential in creating international forums sponsored by public institutions, where entrepreneurial social and professional exchanges are encouraged, can be an effective way for cross-border collaboration between entrepreneurs of different nationalities. As the EG case proved, connecting innovative local entrepreneurs with international institutions can be very effective and fruitful, specifically in knowledge sharing and idea exchange. Great problems such as sustainability need synergies that use ideas from multiple cultures and backgrounds to be practical and effective. International forums, guided by the entrepreneurial spirit, where academia and industry of multiple geographical locations converge and collaborate, can be practical tools for solving global issues. The EntREsilience project is an excellent example of cross-country academic collaboration with international agencies. Such forums can also be a good source of crowdfunding for entrepreneurs helping them scale their operations. Asking entrepreneurs to solve social problems through professional competitions on local, regional and international levels can result in novel solutions and generate more significant and effectual public–private joint ventures.

As stated in the discussion section, sustainability efforts are hampered by the economic foundations that took shape over the past three centuries. Attaching our well-being and quality of life to economic growth has resulted in an unsustainable world posing an existential threat. Considering ecological care as part of business sustainability by accepting nature as a business stakeholder can only change the fundamentals of our economic paradigm. Communities can be a very efficient tool in altering the established economic fundamentals by promoting non-economic-based ideas for a better quality of life through philosophies like the sufficiency economy [9].

**Author Contributions:** Conceptualization, M.H., R.S., E.A. and G.P.M.; writing—original draft preparation, M.H.; writing—review and editing, M.H., R.S., E.A. and G.P.M.; funding acquisition, R.S. and E.A. All authors have read and agreed to the published version of the manuscript.

**Funding:** This research project is supported by UKRI and EPSRC Global Challenges Research Fund (Reference: EP/V208480/1).

**Institutional Review Board Statement:** Not applicable.

**Informed Consent Statement:** Not applicable.

**Data Availability Statement:** Data are available upon reasonable request.

**Conflicts of Interest:** The authors declare no conflict of interest. The funders had no role in the design of the study; in the collection, analyses, or interpretation of data; in the writing of the manuscript; or in the decision to publish the results.

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
