# Peer review of "How Has the COVID-19 Crisis Transformed Entrepreneurs into Sustainable Leaders?"

_sustainability, doi:10.3390/su15065358_

Round 1

Reviewer 1 Report

Title: How Covid-19 Crisis Transformed Entrepreneurs to Sustainable Leaders?

From the content of the work this research reflects more of entresilience but this construct does not reflect anywhere in the topic. Hence, the title should be redrafted and should also adequately contain the independent and dependent constructs.

Keywords: I encourage/suggest that the keywords be arranged in alphabetical order.

Introduction

Research background

You may want to explain properly the difference between the sub-title “introduction” and “research background”. I believe there are no differences. Hence the researchers are to consider merging the sub-titles “Introduction and Research Background”.

This article is obviously a build-up from previous work; line 58 establishes that fact, at this point it becomes confusing to readers because they were not pre-informed about the major themes of the work as entresilience is never a part of the topic. Authors should look into that and give readers clues as to exactly what they have on their minds.

 However, in line 79, the word “during and after” should be turned into small letters.  Also, in lines 79 – 83, the researchers already have them in the introduction hence why repeat the same information?

 In line 99, The process model of entrepreneurial resilience is not well explained. Where was the model gotten from? What`s the relevance of the model to the study?

There are no clearly stated objectives to guide readers in the article. Authors could consider stating the objectives of the study for clarity.

The study does not have a clear conceptual directive, nor does it have any theoretical inclination that projects and gives credence to the present study, more so, there is no empirical evidence to support the argument of the authors. This could be checked out. this would have given a fair understanding to readers.

Lines 111-118 (how were the themes formed?

What informed the choice of these entrepreneurs?

The Extant literature (line 118) consulted was not mentioned.

Lines 123-125: the EntREsilent data analysis that identified five polar pairs of contrasting responses to the pandemic crisis has no background information for proper flow and comprehension.

All variables expressed on lines 127-132 were never discussed in the work.

Methodology:  the methodology did not spell out a clear strategy followed for easy comprehension.

There are no step-by-step explanations of the different issues that should appear at the methodology section;

Population description needed

Why exploratory research design?

why the use of purposive sampling?

Why was the choice of entrepreneurs used? (inclusion and exclusion criteria) etc. authors could do justice to these issues for a clearer understanding.

The analysis and results are consistent with what was done. It’s quite captivating and good. the issue there is that readers who have no previous information on the study may be confused.

The discussion is consistent with the results but could be improved upon.

The communication is highly technical.

Plagiarism: is currently 7% which is highly commendable.

References: The currency of scholarly literature used for the work is average. This is commendable.

Author Response

Thank you for your knowledgeable and insightful comments. The comment proved to be valuable guides in submission revision. 

The article has been revised. The Abstract, Introduction, Research background, theoretical foundation and research questions and Methodology sections have been rewritten to answer your comments. Below are the answers to your respective queries. 

Keywords: I encourage/suggest that the keywords be arranged in alphabetical order.

Keywords have been revised

You may want to explain properly the difference between the sub-title “introduction” and “research background”. I believe there are no differences. Hence the researchers are to consider merging the sub-titles “Introduction and Research Background”.

The sub-titles have been revised to reflect the revised text.

This article is obviously a build-up from previous work; line 58 establishes that fact, at this point it becomes confusing to readers because they were not pre-informed about the major themes of the work as entresilience is never a part of the topic. Authors should look into that and give readers clues as to exactly what they have on their minds.

The revised text explains EntREsilience in greater detail.

 However, in line 79, the word “during and after” should be turned into small letters.  Also, in lines 79 – 83, the researchers already have them in the introduction hence why repeat the same information?

The issue is resolved.

 In line 99, The process model of entrepreneurial resilience is not well explained. Where was the model gotten from? What`s the relevance of the model to the study?

The revised text explains the EntREsilience process model in greater detail.

There are no clearly stated objectives to guide readers in the article. Authors could consider stating the objectives of the study for clarity.

The research background, theoretical foundation and research questions section end with detailed research questions.

The study does not have a clear conceptual directive, nor does it have any theoretical inclination that projects and gives credence to the present study, more so, there is no empirical evidence to support the argument of the authors. This could be checked out. this would have given a fair understanding to readers.

Hopefully, the revised text has resolved this issue.

What informed the choice of these entrepreneurs?

The revised research background and methodology explains the case choices in detail.

The Extant literature (line 118) consulted was not mentioned.

The revised text has more citations to resolve this issue.

Lines 123-125: the EntREsilent data analysis that identified five polar pairs of contrasting responses to the pandemic crisis has no background information for proper flow and comprehension.

The revised text explains the EntREsilience polar pair in detail.

All variables expressed on lines 127-132 were never discussed in the work.

The revised text explains the EntREsilience polar pair in detail.

Methodology:  the methodology did not spell out a clear strategy followed for easy comprehension.

The revised research background and methodology explains the strategy in detail.

There are no step-by-step explanations of the different issues that should appear at the methodology section;

Hopefully, the revised text has resolved this issue.

Population description needed? Why exploratory research design? why the use of purposive sampling? Why was the choice of entrepreneurs used? (inclusion and exclusion criteria) etc. authors could do justice to these issues for a clearer understanding.

The revised research background and methodology explanations aimed at answering the above comments.

The analysis and results are consistent with what was done. It’s quite captivating and good. the issue there is that readers who have no previous information on the study may be confused.

Hopefully the revised text has resolved this issue.

The discussion is consistent with the results but could be improved upon; From the content of the work this research reflects more of entresilience but this construct does not reflect anywhere in the topic. Hence, the title should be redrafted and should also adequately contain the independent and dependent constructs.

Hopefully, the revised text has answered these comments.

Reviewer 2 Report

Thank you for providing the opportunity to review this manuscript titled "How Covid-19 Crisis Transformed Entrepreneurs to Sustainable Leaders?" 

the paper aims to add to a five-country longitudinal study that was launched in 2020 to understand how 14 entrepreneurs manifest resilience in response to the Covid-19 pandemic crisis . While the original project analysed entrepreneurial responses from  China, Malaysia, the Philippines, Thailand and UK , the present research  is limited to Thai context only. The authors need to provide a stronger justification of confining their study to this particular context.

 A better job needs to be done in presenting the theory underlying the resilience process model. You should highlight the key limitations and strengths of existing research on the model and how your study aims to address some of the limitations.  

Introduction can be improved by addressing the following questions: Why should we be interested in your research findings?  Which theoretical lens are you addressing? What is your theoretical contribution?

The paper's arguments re the resilience process model be built on an appropriate base of theory. 

In terms of methodology, the authors should provide some background information on the selected design. I.e. "The Gioia systematic methodology for grounded-theory-based interpretive research" why you have adopted this approach. What was the criteria for selecting the sample? How  the selected methodological approach is  well suited to answer your research questions (what are they?) and in probing the effect of moderators? 

Author Response

Thank you for your knowledgeable and insightful comments. The comment proved to be valuable guides in submission revision. 

The article has been revised. The Abstract, Introduction, Research background, theoretical foundation and research questions and Methodology sections have been rewritten to answer your comments. Below are the answers to your respective queries. 

the paper aims to add to a five-country longitudinal study that was launched in 2020 to understand how 14 entrepreneurs manifest resilience in response to the Covid-19 pandemic crisis . While the original project analysed entrepreneurial responses from  China, Malaysia, the Philippines, Thailand and UK , the present research  is limited to Thai context only. The authors need to provide a stronger justification of confining their study to this particular context.

The revised Research background and Methodology explain the case choices in detail.

 A better job needs to be done in presenting the theory underlying the resilience process model. You should highlight the key limitations and strengths of existing research on the model and how your study aims to address some of the limitations. 

The revised research background explains the theoretical foundation of the study in detail.

Introduction can be improved by addressing the following questions: Why should we be interested in your research findings?  Which theoretical lens are you addressing? What is your theoretical contribution?

The revised introduction and research background are aimed at answering these comments.

The paper's arguments re the resilience process model be built on an appropriate base of theory.

The revised research background aims to clarify the study's theoretical foundation.

In terms of methodology, the authors should provide some background information on the selected design. I.e. "The Gioia systematic methodology for grounded-theory-based interpretive research" why you have adopted this approach. What was the criteria for selecting the sample? How  the selected methodological approach is  well suited to answer your research questions (what are they?) and in probing the effect of moderators?.

The revised research background and methodology explanations are aimed at answering the above comments.

Reviewer 3 Report

How Covid-19 Crisis Transformed Entrepreneurs to Sustainable Leaders?

Abstract

The goal is not expressed in an incisive way, it would be more useful to motivate the goal of enriching the data of the EntREsilience project. What is the purpose of exploratory research? What research question does it answer?

Ineffective and descriptive keywords; the keyword ‘stakeholders’ is  too general

Introduction

The introduction is too aimed at describing an already existing project but does not highlight the research gap that this study intends to fill, does not introduce questions or research hypotheses and does not justify the study's focus on companies in a single country among those included in the project from which it takes place.

Research background

It is reduced to the sole description of the project from which we start. There is a total lack of literature review on the theoretical constructs to which the study should refer (such as stakeholder theory or crisis management)

Methodology

Explain how the 4 Thai cases considered were selected. Why the focus on Thai companies?

The data analysis phases are difficult to follow without having knowledge of the EntREsilience methodology. The authors should describe it, in a concise but exhaustive way, to allow readers to understand what the authors' intervention is in the context of this research and to follow its development.

Findings

They need to be reported in a clearer and easier to read way. The considerations written in this section are not the reading of the findings obtained from the research but are considerations taken from the literature that should be anticipated to the conceptual background. It's not really clear what the search results are.

Discussion

Not having knowledge of the findings, it is difficult to understand the discussion. What is the authors' original consideration of prior knowledge? The authors state “The inductive results of this longitudinal study showed how the awareness of their stakeholder challenges shaped the entrepreneurial response”. However, this thing does not seem to emerge from reading the findings or from their discussion.

Overall, the study appears weak in both theoretical foundations and methodological rigor. The objective is not incisive and the choice of the focus of the analysis is not clear. For all of the above reasons, it is advisable to rethink the research design.

Author Response

Thank you for your knowledgeable and insightful comments. The comment proved to be valuable guides in submission revision. 

The article has been revised. The Abstract, Introduction, Research background, theoretical foundation and research questions and Methodology sections have been rewritten to answer your comments. Below are the answers to your respective queries. 

Abstract

 The goal is not expressed in an incisive way, it would be more useful to motivate the goal of enriching the data of the EntREsilience project. What is the purpose of exploratory research? What research question does it answer?

Hopefully the revised text has answered these comments.

Ineffective and descriptive keywords; the keyword ‘stakeholders’ is  too general

Keywords have been revised

Introduction

The introduction is too aimed at describing an already existing project but does not highlight the research gap that this study intends to fill, does not introduce questions or research hypotheses and does not justify the study's focus on companies in a single country among those included in the project from which it takes place.

Research background

It is reduced to the sole description of the project from which we start. There is a total lack of literature review on the theoretical constructs to which the study should refer (such as stakeholder theory or crisis management)

Methodology

Explain how the 4 Thai cases considered were selected. Why the focus on Thai companies?

The data analysis phases are difficult to follow without having knowledge of the EntREsilience methodology. The authors should describe it, in a concise but exhaustive way, to allow readers to understand what the authors' intervention is in the context of this research and to follow its development.

The revised introduction, research background and methodology explanations aimed at answering the above comments.

Findings

They need to be reported in a clearer and easier to read way. The considerations written in this section are not the reading of the findings obtained from the research but are considerations taken from the literature that should be anticipated to the conceptual background. It's not really clear what the search results are.

Discussion

Not having knowledge of the findings, it is difficult to understand the discussion. What is the authors' original consideration of prior knowledge? The authors state “The inductive results of this longitudinal study showed how the awareness of their stakeholder challenges shaped the entrepreneurial response”. However, this thing does not seem to emerge from reading the findings or from their discussion.

Overall, the study appears weak in both theoretical foundations and methodological rigor. The objective is not incisive and the choice of the focus of the analysis is not clear. For all of the above reasons, it is advisable to rethink the research design.

Hopefully the revised text has answered these comments.

Round 2

Reviewer 3 Report

The revision and integration work carried out by the authors in this new version of the manuscript was important and certainly made the research more understandable, improving its potential impact on the field. However, the conceptual construction is still a bit weak. Authors cite a literature review conducted for the EntREsilience project that led to gaps being defined. Since there is no reference to previous studies showing the results of this literature review, I advise the authors to summarize them in this work. The entire section 2 should be reordered, starting from the theoretical foundations, ie from the stakeholder theory and proposing a summary on the state of the art of academic research on sustainable leadership behavior and on business models for sustainability. Only then, by dividing section 2 into several paragraphs, should the authors present an overview of the EntREsilience project that initiated this manuscript.

Beware of typos, there are several throughout the manuscript. Good luck!

Author Response

Dear Reviewer 3

As advised, we have revised section 2. It summarises the literature review that formed the theoretical base of EntREsilience, stakeholder theory, sustainable business and sustainable leadership.

Thank you for your guidance and suggestions. 

Best regards
